# Changes in Serum Oxytocin Levels under Physiological and Supraphysiological Gonadal Steroid Hormone Conditions in Women of Reproductive Age: A Preliminary Study

**DOI:** 10.3390/nu14245350

**Published:** 2022-12-16

**Authors:** Ayaka Tachibana, Yuri Yamamoto, Hiroki Noguchi, Asuka Takeda, Kou Tamura, Hidenori Aoki, Saki Minato, Maimi Uchishiba, Shota Yamamoto, Shuhei Kamada, Atsuko Yoshida, Riyo Kinouchi, Kanako Yoshida, Takeshi Iwasa

**Affiliations:** 1Department of Obstetrics and Gynecology, Institute of Biomedical Sciences, Tokushima University Graduate School, 3-18-15 Kuramoto-Cho, Tokushima 770-8503, Japan; 2Department of Renal and Genitourinary Surgery, Graduate School of Medicine, Hokkaido University, Sapporo 060-0808, Japan

**Keywords:** oxytocin, estrogen, IVF/ICS

## Abstract

Oxytocin (OT) affects many behavioral, psychological, and physiological functions, including appetite and body weight regulation. Central and peripheral OT levels are markedly affected by gonadal steroids, especially estrogen, and the anorectic effects of estrogen are partially mediated by OT in rodents. In this study, the relationship between the estrogen milieu and serum OT levels was evaluated in women of reproductive age under physiological (*n* = 9) and supraphysiological estrogenic conditions (*n* = 7). Consequently, it was found that serum OT levels were increased in physiological (the ovulatory phase) and supraphysiological (on the day of the human chorionic gonadotropin trigger in an ovarian stimulation cycle) estrogenic conditions, and that serum OT levels were positively correlated with serum estradiol levels. On the other hand, serum OT levels were negatively correlated with serum progesterone levels, and there was no correlation between serum and follicular OT levels. These results suggest that OT levels may be positively and negatively regulated by estrogen and progesterone, respectively, in humans. However, the physiological roles of these actions of gonadal steroids on OT remain unclear.

## 1. Introduction

Oxytocin (OT) is a 9-amino acid neuropeptide, which is synthesized in the hypothalamus and secreted from the posterior lobe of the pituitary gland [1]. The magnocellular oxytocin neurons project into the posterior pituitary gland and secrete OT into the peripheral circulatory system, while the parvocellular oxytocin neurons project and secrete OT into several regions of the central nervous system, such as the arcuate nucleus. It is well established that peripheral OT promotes labor and lactation in mammalian females [2]. Recently, it has been revealed that OT affects behavioral, psychological, and physiological functions, such as trust, bonding, empathy, social communication, metabolism, appetite, and body weight regulation [3,4,5]. In addition, it has been reported that the administration of OT ameliorates some psychiatric and metabolic/nutritional disorders in humans and experimental animals [5,6,7,8].

In our previous studies, it was shown that serum OT, hypothalamic OT, and OT receptor (OTR) mRNA expression levels are markedly affected by gonadal steroid hormones in females. Serum OT, hypothalamic OT, and OTR mRNA levels were decreased in ovariectomized female rats, and concomitant increases in food intake and body weight were seen, and these changes were reversed by estradiol (E2) supplementation [9], indicating that estrogen may positively regulate the central and peripheral OT systems. In addition, exogenous OT administration attenuated food intake and body weight gain, and reduced body fat levels without having any apparent adverse effects in ovariectomized and aged female rats with irregular estrous cycles [5,10]. As estrogen also has anorectic and anti-obesity effects in females, it is assumed that these effects of estrogen may be mediated, at least in part, by stimulation of the OT system. Furthermore, the serum OT levels of female rats’ administered dihydrotestosterone, an androgen, were decreased, and exogenous OT administration attenuated food intake [7]. As androgens have orexigenic and pro-obesity effects in females, it is speculated that these effects of androgens may be induced by attenuation of the OT system [11].

As noted above, it has been suggested that relationships exist between the OT system and the gonadal steroid hormone milieu in female rodents. On the other hand, no previous study has evaluated whether the same mechanism is in operation in humans. In this study, the relationship between the estrogen milieu and serum OT levels was evaluated in women of reproductive age under physiological and supraphysiological estrogenic conditions. In addition, the relationship between serum and follicular OT levels was also evaluated because it has been shown that some brain derived factors flow into follicular fluid and that their concentrations in follicular fluid are sometimes high [12].

## 2. Materials and Methods

### 2.1. Study Design

This prospective cohort study was approved by the clinical research review board of Tokushima University (Approval No. 4021 and 4098). An opportunity to opt-out was provided or written informed consent was obtained from all subjects. Infertile patients (31 to 41 years old) who were scheduled to undergo infertility examinations or IVF/ICSI treatment between July 2020 to May 2022 were enrolled. The primary aim of this study was to evaluate the relationship between serum E2 and OT concentrations. In addition, the relationship between serum P4 and OT concentrations was also evaluated. The secondary aim was to compare the OT levels in serum and follicular fluid.

### 2.2. Changes in Serum OT Concentrations during the Menstrual Cycle

Infertile women with regular menstrual cycles were enrolled in this experiment. Serum samples were obtained in the follicular (menstrual days 3–5), ovulatory (when the follicle size reached ≥ 17 mm), and luteal (5–7 days after ovulation) phases. The serum OT and E2 concentrations were measured in all phases. In addition, the serum P4 concentration was measured in the luteal phase.

### 2.3. Changes in Serum OT Concentrations and Follicular OT Concentrations during Ovarian Stimulation

Infertile women were scheduled for controlled ovarian stimulation, involving either a GnRH agonist- or antagonist-based protocol. Details of the ovarian stimulation protocols were provided in our previous study [13]. Briefly, in the GnRH agonist-long protocol, the patients were started on buserelin acetate in the midluteal phase of the preceding cycle. The administration of a follicle-stimulating hormone (FSH) or a human menopausal gonadotrophin (hMG) was started within seven days of withdrawal bleeding, and the initial dose was adjusted based on the patient’s ovarian reserve, such as antral follicular count, age, and anti-Müllerian hormone level. Administration with initial dose of FSH/hMG was maintained for about five days. Subsequent FSH or hMG doses were determined according to follicular maturation and number, as assessed by transvaginal ultrasound sonography. In the GnRH antagonist-based protocol, ganirelix treatment was started when the dominant follicle reached ≥14 mm in diameter. When the follicle size reached around 18 mm, 5000 IU human chorionic gonadotrophin (hCG) was administered. Around 36 h after the hCG administration, transvaginal ultrasound-guided oocyte pick-up (OPU) was performed. Semen from the participant’s partner was prepared using the swim-up technique, and IVF, ICSI, or both was carried out. Single embryo transfer was conducted at the cleavage stage (day 2 or day 3) or blastocyst stage (day 5 or day 6) based on the fertilized ovum count. As some patients were at a high risk of ovarian hyperstimulation syndrome, a freeze-all strategy was employed, and frozen-thawed embryo transfer was performed. Clinical pregnancy was defined as the presence of at least one intrauterine gestational sac, according to transvaginal sonography. Serum samples were obtained on menstrual days 3–5 in the previous cycle (baseline), on the initial day of stimulation, and on the day of the hCG trigger. Follicular fluid from the first retrieved follicle was collected to avoid contamination by blood or the flushing medium or the mixing of follicular fluid during oocyte retrieval.

### 2.4. Hormone Assay

Serum was separated from blood samples by centrifugation at 3500 rpm for 15 min, and the samples were frozen at −40 °C. Follicular fluid samples were centrifuged at 1500 rpm for 15 min, and the supernatants were stored at −40 °C. Serum samples were sent to a commercial laboratory (ASKA Pharmaceutical Medical Inc., Co., Ltd., Fujisawa City, Japan), where serum OT levels were measured using a chemiluminescent enzyme immunoassay. Serum samples were also sent to another commercial laboratory (SRL, Tokyo, Japan) so that E2 and P4 levels could be measured using an electrochemiluminescence immunoassay.

### 2.5. Statistical Analyses

Data analyses were performed using one-way ANOVA followed by the Tukey–Kramer test or Student’s t-test. As the serum levels of OT vary markedly among individuals, relative values were calculated, i.e., the concentrations seen in the follicular phase or baseline were set as 1.0 to evaluate the patterns of change seen during the menstrual cycle or ovarian stimulation protocol.

## 3. Results

### 3.1. Changes in Serum OT Concentrations during the Menstrual Cycle

A total of nine women with regular menstrual cycles were enrolled in this experiment. Their serum estradiol (E2) concentrations were significantly higher in the ovulatory and luteal phases than in the follicular phase, whereas their serum OT concentrations did not differ among these three phases (Figure 1A,C). On the other hand, their relative serum OT levels were significantly higher in the ovulatory phase than in the follicular phase (Figure 1B), and their relative serum E2 levels were significantly higher in the ovulatory and luteal phases than in the follicular phase (Figure 1D). There was a significant positive correlation between their serum OT and E2 concentrations (Figure 2A), whereas there was a significant negative correlation between their serum OT and progesterone (P4) concentrations (Figure 2B).

### 3.2. Changes in Serum OT and Follicular OT Concentrations during Ovarian Stimulation

A total of seven women were enrolled in this experiment. Six women were treated with the gonadotropin-releasing hormone (GnRH) agonist-based protocol, and one woman was treated with the GnRH antagonist-based protocol. The serum E2 concentration and relative serum E2 concentration on the day of the human chorionic gonadotropin (hCG) trigger were significantly higher than those seen at the baseline and on the initial day of stimulation (Figure 3C,D). The serum OT concentration and relative serum OT concentration on the day of the hCG trigger were significantly higher than those seen at the baseline and on the initial day of stimulation (Figure 3A,B). On the other hand, the serum OT concentration and relative serum OT concentration on the initial day of stimulation did not differ from those observed at the baseline (Figure 3A,B). The OT concentration in follicular fluid was lower than that in serum on the day of the hCG trigger, and there was no correlation between the follicular and serum OT concentrations (Figure 4). In addition, we could not detect any relationship between serum OT and follicular OT levels and the outcome of IVF-ICSI (data not shown), as the sample size may be too small to analyze.

## 4. Discussion

In 1906, it was reported that extracts from human posterior pituitary were able to contract the uterus of a pregnant cat [14,15]. The responsible peptide was later synthesized and sequenced by du Bigneaud in 1953 [16]. OT is a one of the neuropeptides which is produced in parvocellular neurons of the paraventricular nucleus (PVN) and magnocellular neurons of the PVN and supraoptic nuclei in hypothalamus [17]. Most OT from hypothalamic magnocellular neurons is transmitted to the posterior pituitary and secreted to the peripheral level. In addition, in central levels, OT acts as neurotransmitter or neuromodulator in the hippocampus, striatum, suprachiasmatic nucleus, bed nucleus of stria terminalis, and brain stem. Thus, OT acts on both central and peripheral levels and its functions may be clearly distinguished. The OT receptor (OTR) belongs to the G-protein-coupled receptor superfamily and its expression profile is stage- and tissue-specific, e.g., up-regulated in the term uterine myometrium and down-regulated in non-pregnant myometrium. The OTR has been identified in many tissues, including adipocytes, heart, kidney, pancreas, and some brain areas. In addition, the OTR is expressed in the mammary glands and uterine myometrium in females. Main peripheral actions of OT are smooth muscle contraction and lactation. On the other hand, it has been shown that central OT are related in many psychological and physiological functions, such as trust, bonding, empathy, social communication, metabolism, appetite, and body weight regulation.

In this study, we have shown that serum OT levels increased when serum E2 levels were elevated under both physiological and supraphysiological conditions. In addition, we have shown that serum OT levels were positively correlated with serum E2 levels in physiological conditions. As far as we know, there are only a few studies that shows that a relationship exists between OT and gonadal steroid hormone levels in humans [17]. In our previous study, it was shown that the changes in serum OT levels and hypothalamic OT mRNA expression levels were linked in female rats, i.e., when the serum OT level increased, the hypothalamic OT mRNA level also increased [9]. Thus, it can be speculated that central OT levels, as well as serum OT levels, may be increased under high serum E2 conditions in humans. In this study, although we could not detect central OT levels, it may be assumed that central OT levels, as well as serum OT levels, may be increased in the ovulatory phase in physiological conditions and on the day of hCG injection in supra-physiological conditions.

As noted above, OT affects some behavioral, psychological, and physiological functions [3,4,5]. In particular, recent studies have revealed that OT plays pivotal roles in the metabolism, appetite, and body weight regulation systems in animals and humans [8,18]. For example, OTR-deficient mice exhibited obesity due to an increased amount of visceral fat [19], and the central or peripheral administration of OT decreased food intake [8,20,21,22,23,24,25] and promoted lipolysis [21,22,23,26,27,28,29,30,31,32] in rodents, primates, and humans. In a previous study, we have also shown that the 6-day peripheral administration of OT caused marked reductions in body weight gain and food intake in ovariectomized obese rats [5]. In addition, we found that the 6-day peripheral administration of OT caused reductions in visceral and subcutaneous fat weight and adipocyte size. In addition, we have shown that the administration of OT does not give any adverse effects, such as hepatic injury, febrile response, and behavioral abnormality. Similarly, in another study, we have shown that the 12-day peripheral administration of OT caused body weight reduction and adipocyte size in peri- and postmenopausal female rats [10]. Similarly, the 12-day peripheral administration of OT caused a reduction of serum lipid levels, whereas it did not affect the serum level of hepatic enzyme and renal factors, indicating that chronic OT administration may not induce obvious adverse effects. Furthermore, as noted above, we have revealed that hypothalamic OT and OTR mRNA levels and serum OT concentration are decreased in ovariectomized female rats and that chronic supplementation of E2 recovers these alterations in previous study [9]. These findings indicate that suppressive effects of E2 on appetite and adiposity may be mediated by OT and that OT can be used to treat or prevent menopause-induced metabolic disorders, as it does not have any adverse effects. In addition, previous studies have shown that estrogen increases the levels of central and peripheral anorexigenic factors, while it reduces the levels of orexigenic factors [33,34,35,36], and that these changes become prominent in the estrus and proestrus stages, when serum estrogen levels are elevated [33,34]. Concretely, the hypothalamic expression of orexigenic factors, agouti-related protein (AgRP), and neuropeptide Y (NPY) are decreased at the estrous stage, where serum estrogen level is increased and body weight and food intake are decreased, in female mice [33]. These cyclic alterations of food intake are abolished, and exogenous E2-induced reduction of food intake is not observed in AgRP degenerated mice, indicating that these factors are indispensable for the changes of body weight regulation during estrous cyclicity. Conversely, the serum leptin level, which is one of the potent anorexigenic factors, is increased in the hyperestrogenic proestrous stage in female rats, concomitantly with the increase of leptin mRNA expression in adipose tissue, indicating that these alterations may also play a role in regulating appetite during estrous cycle [34]. It may be that these behavioral changes aim to prioritize reproduction over appetite during the period when the possibility of conception is high. It is possible that the increase in OT levels seen in the ovulatory phase in hyperestrogenic conditions in the present study may also be intended to promote sexual behavior by inhibiting appetite and feeding behaviors. Another possibility is that this increase in OT levels may have psychological effects that promote sexual behavior. OT has been implicated in the promotion of bonding and mating in human and sexual behavior, such as lordosis, and sexual satiety in rodents [37]. For example, past studies using fMRI have revealed the activation of OT receptor-rich brain areas after orgasm or visual stimuli in humans. Similarly, many studies have indicated an increase of serum OT concentrations during sexual arousal and after orgasm. In female rats, OT is known to activate lordosis behavior, i.e., central administration of OT facilitates lordosis and the co-administration of the OT receptor antagonist reduces such OT-induced lordosis. In addition, it has been shown that couples with a greater amount of partner support have high serum OT levels and that serum OT levels are increased by warm partner contact. Taken together, the increase in OT levels seen during the ovulatory phase and in hyperestrogenic conditions may help to coordinate and integrate various physiological and psychological functions to increase the probability of conception. However, because no outcomes related to such behavioral changes could be evaluated in this study, further examinations would be needed to clarify this hypothesis. On the other hand, a negative correlation between serum P4 and OT levels was noted in the luteal phase in this study. As far as we know, this is the first study to show that a relationship exists between P4 and OT in non-pregnant conditions. As mentioned above, reproductive function and appetite/body weight regulation systems are closely linked, i.e., estrogens suppress food intake and body weight gain, whereas progesterone and androgens increase them. Although a physiological dose of progesterone itself does not affect feeding behavior in ovariectomized female rats, progesterone does stimulate appetite and promote body weight gain in the presence of estrogens [38]. Similarly, it has been shown that food intake is lowest during the periovulatory phase, when estrogen levels are high and progesterone levels are low, whereas food intake is increased during the luteal phase, when progesterone levels are high, in the case of women. Thus, it can be speculated that progesterone may suppress the stimulatory actions of estrogens on OT and that these effects may promote appetite during luteal phase in women. Interestingly, it has been reported that OT also affects the progesterone release in some species. OT stimulates progesterone release in a dose dependent manner and the co-administration of OT antagonist blocks such stimulatory effects of OT in bovine corpus luteum under in vitro condition [39]. In addition, OTR is expressed in an immortalized human granulosa-lutein cell line and OT enhanced the effects of forskolin on progesterone production. These previous studies and our present data indicate that progesterone and OT interact with each other, namely, that OT stimulates the production and release of progesterone, whereas progesterone may suppress OT levels. However, the physiological implications of this relationship remains unclear, and further studies are needed to clarify this.

It has been proposed that the composition of follicular fluid strongly influences oocyte quality, developmental competence, and the quality of embryos. For this reason, many studies have focused on the follicular fluid as an important source of potential non-invasive biomarkers of oocyte and embryo quality, as well as biomarkers for predicting clinical outcomes. However, it has not been evaluated whether OT exists in follicular fluid and their associations with serum OT levels. Thus, in this study, we evaluated the OT concentration in follicular fluid during an in vitro fertilization/intracytoplasmic sperm injection (IVF/ICSI) cycle. As a result, it was found that the follicular fluid OT level was much lower than the serum OT level. In addition, there was no significant correlation between serum and follicular OT levels. Previous studies have shown that some brain-derived factors affect follicular growth and egg quality in infertile women. For example, melatonin, which is produced in the pineal gland, is present in follicular fluid and protects oocytes from oxidative damage [40]. Melatonin also has some beneficial effects on oocyte maturation, fertilization, and embryo development [40]. Another study showed that the melatonin level in follicular fluid is higher than that in serum and that follicular fluid and serum melatonin levels are positively correlated in humans [41]. Similarly, we have evaluated the follicular and serum levels of biotin, which is a water-soluble vitamin (B_7_) and is involved in in vivo carbon dioxide fixation reactions as a coenzyme, in a previous study. As a result, we showed that biotin could be detected in follicular fluid and that serum biotin levels and follicular fluid biotin levels were positively correlated [13], indicating that biotin may be taken up into the follicular fluid from blood. On the other hand, as noted above, the follicular OT level was extremely low and not correlated with the serum OT level, indicating that OT may not have marked effects on egg quality.

The limitations of this study are its small sample size and lack of a control group. As to the latter limitation, the differences between males and females and between women of a reproductive age and menopausal women would be needed in future examinations. A further limitation of this study is that the relationship between follicular OT and egg quality, maturation, and fertility rate could not be fully evaluated. Thus, we cannot exclude the possibility that follicular OT affects some actions on the egg. In this regard, further examination would be needed. Another limitation is that we could not evaluate the changes in food intake and body weight during menstrual cycles and ovarian stimulation. Thus, it remains unclear whether changes of serum OT level observed in this study really affect the appetite and weight regulation in these women. As noted above, some studies indicate that OT gives favorable effects on the body weight regulation and metabolism in human and experimental animals. However, most studies focused on the effects of OT in men and male. For example, ablation of OT neurons induces an increase in body weight gain, and the administration of the OT antagonist increases food intake in male mice [42] On the other hand, these changes due to the ablation of OT on body weight are not evident in female mice [42]. Similarly, it has been shown that OT administration inhibits food intake in obese men [25] and that OT treatment decreases abdominal and subcutaneous fat in high-fat-diet induced male mice [29]. We assume that the effects of OT administration on metabolic and feeding behavior are more complex in women of reproductive age, as gonadal hormonal status, i.e., estrogen and progesterone levels, might affects not only endogenous but also exogenous OT actions. Thus, when we evaluate the effects of endogenous and administered OT on psychological and physiological functions in females, we should keep in mind that changes of gonadal steroids during the menstrual cycle (human) and estrous cyclicity (some experimental animals) may affect the result of the examinations. In addition, these perspectives might play important role in the evaluation of the sexual dimorphism of the neuroendocrine functions.

In summary, we showed that serum OT levels were increased when serum E2 levels were elevated under both physiological and supraphysiological estrogenic conditions. It is suggested that these changes may affect various physiological and psychological functions in order to promote sexual behavior and increase the likelihood of conception. On the other hand, there was a negative correlation between serum OT and P4 levels in the luteal phase; however, its physiological implications remain unclear. In addition, the OT level in follicular fluid was extremely low and was not correlated with the serum OT level, indicating that OT may not directly affect ovarian functions.

## Figures and Tables

**Figure 1 nutrients-14-05350-f001:**
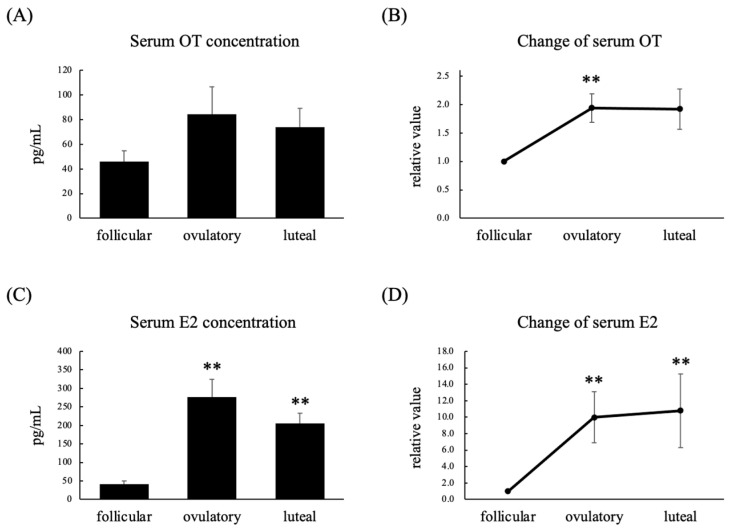
Serum OT concentration (**A**), relative serum OT concentration (**B**), serum E2 concentration (**C**), and relative serum E2 concentration (**D**) in the follicular, ovulatory, and luteal phases in women with regular menstrual cycles. The relative values for each phase were calculated by dividing the observed levels by those seen in the follicular phase. Data are expressed as the mean ± standard error. *n* = 9 per group; ** *p* < 0.01 vs. follicular phase.

**Figure 2 nutrients-14-05350-f002:**
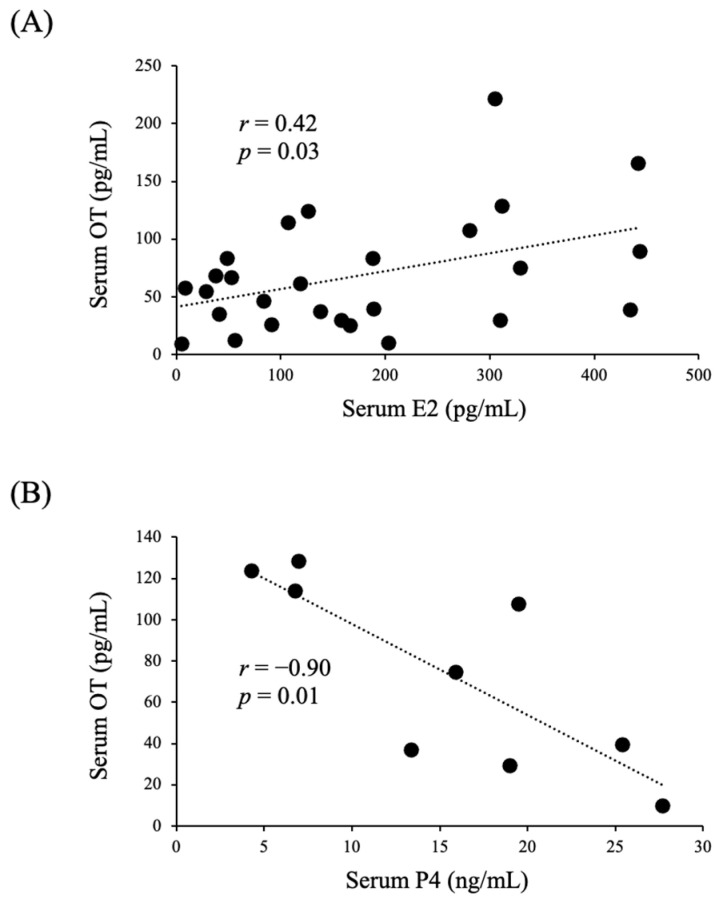
Correlations between serum OT and serum E2 (**A**) or P4 (**B**) levels in women with regular menstrual cycles.

**Figure 3 nutrients-14-05350-f003:**
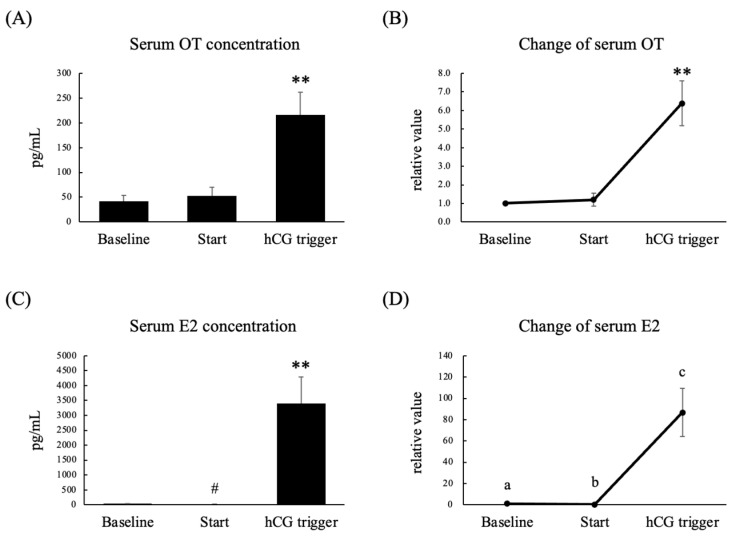
Serum OT concentration (**A**), relative serum OT concentration (**B**), serum E2 concentration (**C**), and relative serum E2 concentration (**D**) on menstrual days 3–5 in the previous cycle (baseline), on the initial day of stimulation (Start), and on the day of the hCG trigger in infertile women who were scheduled for controlled ovarian stimulation. The relative values for each phase were calculated by dividing the measured levels by the levels seen in the follicular phase. Data are expressed as the mean ± standard error. Different letters (a–c) indicate significant differences. *n* = 7 per group; ** *p* < 0.01, ^#^
*p* < 0.05 vs. baseline.

**Figure 4 nutrients-14-05350-f004:**
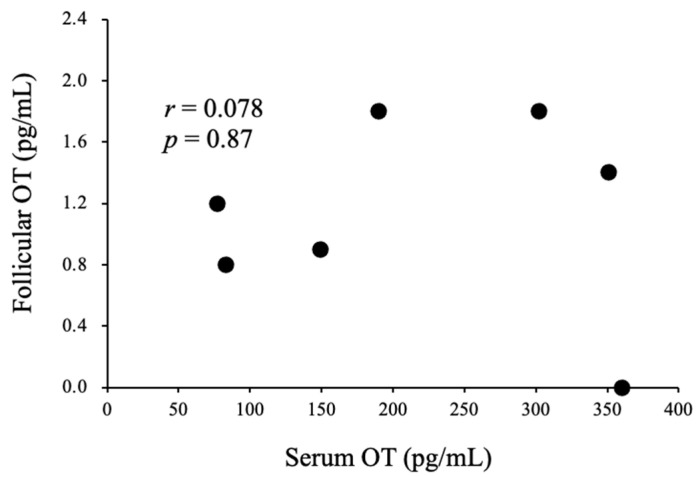
Correlations between serum OT levels on the day of the hCG trigger and follicular OT levels in infertile women during IVF/ICSI cycles.

## Data Availability

Not applicable.

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
