# Peer review of "Changes in Serum Oxytocin Levels under Physiological and Supraphysiological Gonadal Steroid Hormone Conditions in Women of Reproductive Age: A Preliminary Study"

_nutrients, 2022, doi:10.3390/nu14245350_

Round 1

Reviewer 1 Report

This report describes absolute and relative values of peripheral sex steroids in relation to peripheral and follicuular fluid concentrations of oxytocin in "infertile" women undergoing ovulation stimulation. The authors claim that this is the first study to examine this association in humans across the menstrual cycle, although a 2019 meta-analysis by Engel et al in Frontiers Neuroendocrinology suggests otherwise. The authors logically speculate about the relevance of oxytocin's changing secretory pattern in relation to changes in corresponding sexual and eating behaviors over the cycle, but no outcomes related to such behaviors are documented. 

Given the very small sample size, the lack of a "control group", and the use of somewhat imprecise commercial assays ( vs chemiluminescence) to determine hormone measures, it is unclear how these findings move the field forward.

Author Response

I really appreciate for your valuable comments. We revised our manuscript in accordance with your comments as below. We believe that our manuscript has been improved compared with previous version.

  1. As you pointed out, there are a few papers showing the relationship between sex hormone and oxytocin. Thus the sentence of "this is the first study to examine this association in humans across the menstrual cycle" to "there are only a few studies that shows a relationship exists between OT and gonadal steroid hormone levels in humans".
  2. As you pointed out, we did not show the data consolidating our hypothesis. Thus, explanation as follow is added. "On the other hand, no outcomes related to such behavioral changes could evaluated in this study, further examinations would be needed to clarify this hypothesis."
  3. As you pointed out, sample size is small and there is no control group in this study. Thus, these points are explained as limitation as follow. "Limitation of this study is small sample size and lack of a control group. As to the latter limitation, the differences between male and female and women with reproductive age and menopausal women would be needed in future examinations."

Reviewer 2 Report

The author reported the relationship between the estrogen milieu and serum OT levels.

In abstract, “On the other hand, serum OT levels were negatively correlated with serum progesterone levels, and 24 there was no correlation between serum and follicular OT levels.” is result of this study. You should state conclusion of this study.

In introduction, you should add why you evaluated the relationship between serum OT levels and follicular OT levels. What is the significance of follicular OT ?  You should add the information about follicular OT levels.

In Discussion, Line 178-181: You just write method and result in this part. You have to discuss about significance of the relationship between serum OT levels and follicular OT levels above mentioned. You should write clinical significance you can obtain in this study and limitation of this study.

Author Response

I really appreciate for your valuable comments. We revised our manuscript in accordance with your comments as below. We believe that our manuscript has been improved compared with previous version.

  1. In accordance with your comment, configuration of Abstract is changed. In revised version, the conclusion is show as follow. "The increase in OT levels seen around ovulation may promote sexual behavior and inhibit appe-tite and feeding behavior to increase the probability of conception. On the other hand, the physio-logical implications of negative correlation between progesterone and OT levels remain unclear."
  2. In accordance with your comment, the reason of evaluation of relationship between serum and follicular OT was added as follow. " the relationship between serum and follicular OT levels was also evaluated because it has been shown that some brain derived factors affect the follicular growth, maturation and fertility rate in past studies."
  3. In previous version, we made mistake of the composition of paragraph. In revised manuscript, next paragraph (discussion about the serum OT)  is connected.

Round 2

Reviewer 1 Report

Despite clear suggestions for improvements in the manuscript by the reviewers, the revisions were minimal. Typically recommendations made by reviewers are acknowledged in the manuscript, not just in the responses. Additional revisions that are needed include

- acknowledgement in the text of the existing meta-analysis cited by the reviewer and modifcation of  the references. 

- modification of the abstract's conclusions to exclude speculations about cause-effect mechanisms that were not examined in the study, The following text cannot be inferred by the study design or findings: "These results suggest that OT levels are positively regulated by estrogen in humans. The increase in OT levels seen around ovulation may promote sexual behavior and inhibit appetite and feeding behavior to increase the probability of conception". ..

ALso, the sample size of 9 patients should be included in the abstract.

If this was a secondary report of additional findings obtained as part of a larger study, this too should be acknowledged. 

Author Response

Thank you again for your valuable comments. In accordance with your comments, we revised our manuscript. We believe that revised version may be improved due to your advise.

  1. Suggested manuscript is acknowledged in the body of manuscript (line 178, reference number 14) and shown in the reference list (line 347-348).
  2. Our speculation is excluded from Abstract in revised manuscript. Instead, following explanation is added. "These results suggest that OT levels may be positively and negatively regulated by estrogen and progesterone, respectively, in humans. However, the physiological roles of these actions of gonadal steroids on OT remain unclear" (line 23-25).
  3. Sample sizes (physiological : n=9, supra physiological : n=7) are added in revised manuscript (line 17 and 18).
  4. This is not the secondary report.

Reviewer 2 Report

This manuscript was improved from some point. But I think some point should be improved.

 You added as follows "In addition, the relationship between serum and follicular OT levels was also evaluated because it has been shown that some brain derived factors affect the follicular growth, maturation and fertility rate in past studies." 

However, you do not evaluate the follicular growth, maturation and fertility rate in this study from the relationship between serum and follicular OT levels. So your introduction is not accurate. You should include this information to limitaion of this study.

You wrote as follows,"On the other hand, as noted above, the follicular OT level was extremely low  and not correlated with the serum OT level, indicating that OT may not have marked effects on egg quality. " 

However, you discuss about melatonin, you should state about the relationship between egg qualilty and OT leves in both serum and follicle. Because in this study you do not evaluate the relation ship between egg quality and OT levels. This is limitaion of this study. You should add this imformaion in limitation.

Author Response

I really appreciate for your valuable comments. We revised our manuscript in accordance with your comments as below. We believe that our manuscript has been improved compared with previous version.

  1. As you pointed out, we did not evaluate the follicular growth, maturation and fertility rate in this study. Thus, explanation in Introduction is replaced with following one "In addition, the relationship between serum and follicular OT levels was also evaluated because it has been shown that some brain derived factors flow into follicular fluid and that their concentrations in follicular fluid are sometimes high" (line 69-71).
  2. As noted above, we did not evaluate the relationship between follicular OT and egg quality in this study, thus following limitation is show in revised manuscript. "Another limitation of this study is that the relationship between follicular OT and egg quality, maturation and fertility rate could not be evaluated. Thus, we cannot exclude the possibility that follicular OT affects some actions on egg. In this regards, further examination would be needed" (line 284-288).